# Emerging Roles of Endocannabinoids as Key Lipid Mediators for a Successful Pregnancy

**DOI:** 10.3390/ijms24065220

**Published:** 2023-03-09

**Authors:** Alessandro Rava, Viviana Trezza

**Affiliations:** Department of Science, Roma Tre University, 00146 Roma, Italy

**Keywords:** *Cannabis*, endocannabinoids, pregnancy, maternal–fetal interface, gestational inflammation, neurodevelopment

## Abstract

In recent years, *Cannabis* use/misuse for treating pregnancy-related symptoms and other chronic conditions has increased among pregnant women, favored by decriminalization and/or legalization of its recreational uses in addition to its easy accessibility. However, there is evidence that prenatal *Cannabis* exposure might have adverse consequences on pregnancy progression and a deleterious impact on proper neurodevelopmental trajectories in the offspring. Maternal *Cannabis* use could interfere with the complex and finely controlled role performed by the endocannabinoid system in reproductive physiology, impairing multiple gestational processes from blastocyst implantation to parturition, with long-lasting intergenerational effects. In this review, we discuss current clinical and preclinical evidence regarding the role of endocannabinoids in development, function, and immunity of the maternal–fetal interface, focusing on the impact of *Cannabis* constituents on each of these gestational processes. We also discuss the intrinsic limitations of the available studies and the future perspectives in this challenging research field.

## 1. Introduction

*Cannabis* sativa is one of the most commonly consumed illicit drugs worldwide, even among pregnant women [1,2,3,4,5]. In particular, the most recent epidemiological data report an alarming increase in the use of *Cannabis*-derived products among pregnant women both in North America and in the European Union (EU) [1,2,4]. This trend is expected to rise over the next decades due to the progressive legalization and depenalization of *Cannabis* for recreational uses in many Western countries [1,2,3,4]. For instance, since 2012, many US jurisdictions have approved and formalized commercial models of non-medical *Cannabis*, leading to increased accessibility to these products [5,6]. As a consequence, a growing number of people, especially those suffering from chronic pathological conditions, tend to use *Cannabis* as a substitute for one or more drug prescriptions, such as opioids, anxiolytics, and antidepressants [7]. In particular, a recent epidemiological study in the US has reported a high prevalence of *Cannabis* use among pregnant women with disabilities, particularly those with sensory and cognitive deficits [8]. Pregnant women might also use *Cannabis* to relieve nausea, pain, stress, appetite changes, and anxiety during pregnancy [9], a behavior encouraged by the misconception that *Cannabis* use does not lead to detrimental effects on health [10], and likely exacerbated by the distress condition imposed by the COVID-19 pandemic, as recently suggested [11].

Moreover, prenatal *Cannabis* use may be associated with the concomitant use of other drugs of abuse, including tobacco and alcohol, interfering with their detrimental effects [8,12]. In fact, a higher prevalence and frequency of *Cannabis* use have been reported in pregnant women with concurrent opioid use disorder and alcohol consumption [13], highlighting the need to evaluate with more attention the impact of *Cannabis* and polysubstance use on pregnancy outcomes and the underlying pathogenic mechanisms [14].

Conversely, the EU has adopted a more restrictive policy [15,16,17]. Currently, a limited number of *Cannabis*-based medicinal products has been authorized for marketing in several EU member countries to treat specific medical conditions, including those related to chemotherapy, chronic pain, epilepsy, and multiple sclerosis [16,18]. No European government has legalized *Cannabis* sale for recreational use, although several European countries, including Austria, the Netherlands and Portugal, have decriminalized the possession of the drug in quantities which do not exceed that required for average individual consumption [19].

The term “*Cannabis*” is used to define the products derived from *Cannabis sativa*, an annual dioecious plant with complex phytochemistry, including sugars, hydrocarbons, flavonoids, terpenoids, sterols, and more than 110 phytocannabinoids identified up to now [20]. Among them, the most studied phytocannabinoid is Δ9-tetrahydrocannabinol (Δ9-THC), which is responsible of the main psychoactive effects of *Cannabis* [20]. In addition to Δ9-THC, several other non-psychotropic cannabinoids have been identified in *Cannabis sativa*, including cannabidiol (CBD), which in recent years has attracted great interest by clinicians and researchers [21].

Pregnant women are exposed to *Cannabis* through different routes of administration, with smoking being the most common, followed by edible forms and lotions, each with a specific pharmacokinetic profile [22]. Independently from the route of consumption, phytocannabinoids can easily cross the blood-tissue barriers in mammals’ bodies due to their lipophilic nature, and they can impact both male and female reproductive functions. For example, phytocannabinoids can cross the blood-testis barrier and may affect male gonad functions and spermatogenesis, reducing male fertility [23]. Even more interestingly, *Cannabis* constituents are also able to cross the placenta and the blood–brain barrier (BBB). In fetuses and neonates exposed to *Cannabis* preparations during pregnancy and/or lactation, Δ9-THC or its metabolites can be detected in different specimens, such as hair, urine, meconium, and more recently, in the umbilical cord [24]. In rats, Δ9-THC has been revealed in the fetal plasma and brain at approximately 10% to over 30% of the concentrations found in the maternal blood, depending on the timing, dose, and route of exposure [25,26]. The vertical transmission to the fetus of these phytochemicals might explain the growing evidence that associates maternal *Cannabis* use with pregnancy complications and long-term adverse neurological and behavioral effects in the offspring, with an increased risk of psychopathology [27,28,29,30]. This is a worrying issue given the increased concentration of Δ9-THC and other cannabinoids often found today in several illicit *Cannabis* preparations [31,32,33,34].

In this context, a better understanding of the physiological role of the endocannabinoid system for a successful pregnancy and of the impact of prenatal *Cannabis* exposure on the health and well-being of both mother and offspring is of urgent need.

The aim of this review is to address and discuss the main clinical and preclinical literature on the role of endocannabinoids on development, function, and immunity of the maternal–fetal interface, focusing where possible on the impact of *Cannabis* constituents on each of these gestational processes. We summarize the most recent advances in the field providing hints for future research.

## 2. Formation and Development of the Maternal–Fetal Interface: An Overview

Most mammals have adopted a reproductive strategy based on the hemochorial placenta, a transient extraembryonic vascularized organ that mediates nutrient and gas supply from the mother to the developing fetus during gestation [35]. Placenta development, positional, and functional defects are among the most frequent pregnancy complications in mammals, and they can lead to adverse consequences both for maternal and fetal health [36]. Indeed, the placenta regulates multiple crucial processes for embryo development, such as nutritional, excretory, endocrine, and immunological functions, protecting the fetus from chemical and biological insults [37]. These functions are largely conserved among eutherians, despite morphological differences in placenta architecture, highlighting the impact of a dysfunctional placenta on proper fetal growth [38,39].

Most information about the physiological role of the maternal–fetal interface, and the pathological implications arising from its atypical functioning, derives from studies performed on laboratory rodents.

In rodents, the maternal–fetal interface is formed by the maternal decidua, the junctional zone, which constitutes the main endocrine compartment of the fetal placenta, and the complex region of labyrinth, where placental exchange takes place thanks to specialized trophoblasts [40,41] (schematic view in Figure 1). Briefly, after the first blastocyst interactions with the endometrium around Gestational Day (GD) 4.5, the mouse uterine artery branches into several spiral arteries by angiogenesis, and the endometrial stromal cells transform into decidual cells (decidualization) at the site of placentation under the influence of steroid hormones. These processes lead to a substantial tissue and hemodynamic remodelling of the endometrium, a prerequisite needed for pregnancy progression and to allow the bidirectional exchange of micronutrients and gases between oxygenated maternal blood flow and the developing fetus [40]. At the same time, the polar trophectoderm of blastocyst, which resides over the inner cell mass (ICM), differentiates in the ectoplacental cone and in the extraembryonic trophectoderm (GD 5.0–6.5). The first tissue interfaces with the implantation site in the maternal decidua and, in turn, gives rise to trophoblasts that help the embryo anchor and invade the host’s receptive endometrium, and trophoblast giant cells, spongiotrophoblasts, and glycogen trophoblasts forming the junctional zone, whereas the extraembryonic trophectoderm differentiates in chorionic ectoderm, which later fuses with mesoderm-derived allantois. Other embryonic and extraembryonic tissues are generated during gastrulation establishing the three definitive germ layers for future organogenesis [42]. Subsequently, the chorioallantoic fusion and the invagination of allantoic blood vessels into the chorionic plate stimulate the cytotrophoblast progenitors to differentiate and fuse to form the syncytiotrophoblast layer, which structurally supports the formation of the highly vascularized labyrinth, a variation of the human villous placenta [43] (Figure 1). As the pregnancy progresses, the mouse placenta acquires its definitive disc-like architecture, around mid-gestation (GD 10–14.5), becoming the only exchange system capable to respond to the bioenergetic demands of the developing fetus, to which it is connected to via the umbilical cord.

This sequelae of events in rodents leads to the formation of a transient extraembryonic organ that shares many functional characteristics with human placenta, despite species-specific differences exist, making it a valid biological model to identify the molecular mechanisms underlying the intricate crosstalk between mother and fetus both under physiological and pathological conditions [44].

Unlike rodents, which have a hemotrichorial placenta in which two syncytiotrophoblast layers and a third endothelial layer separate maternal circulation from the fetal compartment [45], humans have a hemomonochorial placenta with a single layer of contiguous multinucleated syncytiotrophoblasts that lines the outermost surface of the fetal villous trees [43]. Cytotrophoblast progenitor cells localize below the syncytiotrophoblast layer, where they can differentiate to replenish it or generate extravillous trophoblasts (EVT). By invading decidua, EVTs remodel spiral arteries favoring a sustained maternal blood flow to the placenta at the end of the first trimester of pregnancy [46]. When maternal–fetal blood interface is defined, the risk for the placenta and fetus of a hematogenous transmission of drugs, toxicants, and pathogens circulating in the maternal bloodstream dramatically increases despite the syncytiotroblast layer acts as a selective barrier [47].

A more detailed description of the development and organization of these placental structures and the relative differences between humans and rodents can be found in recent literature (e.g., see [40,45,47]).

## 3. The Endocannabinoid System

Endocannabinoids are a group of endogenous lipid mediators synthetized from membrane phospholipids in response to tissue demands in almost all tissues and body districts [48]. These lipid mediators, together with cannabinoid receptors and metabolic enzymes, constitute the endocannabinoid system (ECS). N-arachidonoylethanolamine (known as anandamide, AEA) and 2-arachidonoylglycerol (2-AG) are the best-studied members of this modulatory system. AEA is synthetized from membrane phospholipid precursors mainly by the sequential action of N-acyltransferase (NAT) and N-acylphosphatidylethanolamine (NAPE)-specific phospholipase d-like hydrolase (NAPE-PLD), and it is hydrolyzed by the fatty acid amide hydrolase (FAAH) to ethanolamine and arachidonic acid, respectively [49]. Conversely, diacylglycerol lipase α (DAGLα) and DAGLβ catalyze the biosynthesis of 2-AG, which is mainly degraded by monoacylglycerol lipase (MAGL) and α/β hydrolase domain containing 2 (ABHD2), ABHD6, and ABHD12 to glycerol and arachidonic acid [50]. Interestingly, both AEA and 2-AG can also be synthetized and metabolized by alternative pathways, which require different intermediates and metabolic enzymes, including cyclooxygenase-2 (COX-2) and lipoxygenases (LOXs) [51]. The relevance of each metabolic pathway might change depending on cell lineage, tissue, developmental stage, and physiological or pathological conditions. For instance, COX-2-mediated oxidation of both AEA and 2-AG to prostaglandins is a key regulator of decidual remodeling [52], and it is increased during inflammatory processes [53].

Once synthetized, endocannabinoids activate specific receptors, modulating a wide range of physiological processes [54,55,56]. Their major and best-characterized targets are the type-1 (CB1) and type-2 (CB2) G protein-coupled cannabinoid receptors. CB1 receptors are highly expressed in the nervous system [57] despite their expression reported in other peripheral organs, including the reproductive tissues [58]. On the contrary, CB2 receptors are primarily expressed in immune cells, where they exert immunomodulatory functions [59,60].

Finally, besides CB1 and CB2 receptors, cannabinoids can signal trough orphan G-protein-coupled receptors (e.g., GPR55 and GPR119), the transient receptor potential vanilloid 1 (TRPV1) channel, and the nuclear peroxisome proliferator activated receptors (PPARs), adding further complexity to the pharmacology of this modulatory system [61].

## 4. The ECS: A Modulatory System for a Successful Pregnancy

### 4.1. Role of Endocannabinoids in the Early Gestational Processes

Alterations of the ECS have been implicated in the etiology of several neurological and neuropsychiatric disorders both in humans and in preclinical models [60,61,62,63]. However, besides a well-documented role of ECS in the modulation of cognitive and emotional processes [64,65], increasing evidence over the last decades suggests that endocannabinoids are also involved in many aspects of male and female reproduction and fertility [23,66]. Endocannabinoids, cannabinoid receptors, and/or metabolic enzymes required for their synthesis and degradation have been identified in many human and rodent reproductive structures and biofluids, as testis, seminal fluid, follicular fluid, ovary, fallopian tube, oviductal fluid, myometrium, endometrium, decidua, placenta, and embryo [58,67,68,69,70,71,72,73,74,75,76,77]. Among these, the female endometrium is a regenerative tissue which undergoes to a profound remodeling during the menstrual cycle, largely under hormonal control [78]. In particular, the transition from the late follicular phase to the early secretory stage of the menstrual cycle, during which ovulation occurs, is characterized by a drastic reduction in plasma level of estradiol and by a significant increment of progesterone, whose level remains elevated up to mid-secretory stage [78]. Interestingly, plasma AEA levels have been correlated with the dynamic changes of sex steroid hormones during the menstrual cycle [79,80], suggesting that these might partially regulate fluctuations in the AEA tone over this period [81,82,83]. In effect, plasma AEA levels are lower in the luteal secretory phase than those in the follicular proliferative phase of the menstrual cycle in healthy women [79,80,82,84]. The endometrium becomes spontaneously receptive during the mid-secretory phase of the menstrual cycle (app. 19–24 days, window of implantation), when AEA levels are low, independently from fertilization [85]. This evidence suggests that low levels of AEA during early pregnancy might be needed to assist early gestational stages, modulating endometrium receptivity to blastocyst implantation and development. The spatiotemporal regulation of AEA tone during the menstrual cycle and across pregnancy requires that the activity of AEA-metabolic enzymes, NAPE-PLD and FAAH, is tightly controlled to create the appropriate conditions for placenta development [77,86]. This fine-tune regulation is particularly evident in rodents, where decidualization does not begin until blastocyst attachment to the endometrial surface takes place [87]. At this time, implantation sites present low levels of AEA, as determined by a reduced NAPE-PLD/FAAH ratio, in contrast to the adjacent inter-implantation sites, where high AEA concentrations have been reported [77,86,88,89].

On the other hand, a sustained AEA signaling increases the uterine refractoriness to embryo implantation [89,90], and inhibits the formation of the maternal decidua impairing endometrial stromal cell survival and their differentiation in decidual cells [91]. Indeed, an increased AEA tone impairs both the oviductal embryo transport and implantation processes in rodents by CB1-dependent mechanisms [90,92]. Accordingly, genetic or pharmacological inhibition of CB1 receptors rescued the impaired blastocyst development and oviductal retention induced by plant-derived cannabinoids [90,93]. Similarly, the synthetic cannabinoid agonists WIN55,212 and CP55,940 were found to negatively modulate the preimplantation development of blastocysts [94]. Overall, these findings suggest that CB-mediated endocannabinoid signaling contributes to the crosstalk between blastocyst and maternal tissues underlying oviductal transport and uterine implantation dynamics. In line with these observations, Li and coworkers recently found that CB1 and CB2 knockout mice showed a higher incidence of pregnancy failure and implantation defects, primarily due to improper blastocyst-endometrium interaction [95]. Indeed, although the authors found an on-time implantation in double knockouts, in contrast to previous works [93,96], the luminal epithelial organization of the endometrium at the implantation chamber appeared abnormal and associated with increased edema compared to controls [95].

In addition, CB1-mediated signaling regulates the oviductal transport of the embryo to the uterus acting on the adrenergic system, which in turn, controls the coordinated oviductal muscle contractility and relaxation [96]. An excessive local AEA tone might impair oviductal smooth muscle activity, which is essential for the embryo passage from the ampulla to the uterus, a condition that could favor ectopic pregnancies in humans. In humans, ectopic pregnancy has been associated with high AEA levels and deregulated CB1 and FAAH expression/activity in the fallopian tubes [71,97] and the peripheral blood [98]. Furthermore, genetic variations of the Cnr1 gene were found in pregnant women suffering from ectopic pregnancy [97] and preeclampsia [99], suggesting that CB1 dysregulations may be a potential risk factor for gestational complications in humans, despite some contradictory results among studies have been reported [100]. For example, some authors reported an increased placental CB1 expression, especially in the syncytiotrophoblast layer, in patients suffering from preeclampsia compared to healthy women [101], while others reported no differences [102]. Of interest, Lombó and colleagues more recently demonstrated an upregulation of CB1 expression within the chorionic villi in preeclamptic patients [103]. The same authors associated CB1 upregulation with increased collagen deposition and lipid peroxidation in these placental compartments, providing a potential role for the ECS, and especially for CB1 receptors, in preeclampsia [103]. Due to these controversial findings, the role of CB1 and other components of the ECS in the pathogenesis of preeclampsia remains to be fully elucidated.

Finally, it is worth noting that implantation and placentation failures are not associated exclusively with maternal defects. Appropriate AEA signaling on the fetal blastocyst is essential not only for uterine receptivity, but also for promoting embryo implantation competency [93,104]. Endocannabinoid signaling also impacts multiple biological pathways in preimplantation blastocysts, including those related to trophoblast cell migration and mobility [105]. Indeed, both CB1^-/-^ and FAAH^-/-^ mouse blastocysts showed in vitro compromised trophoblast cell migration compared to WT counterparts [105]. Considering that trophoblast invasion is a key step for proper implantation, a reduced mobility of these cells due to ECS dysregulations might compromise the correct blastocyst trophectoderm infiltration in the decidualized endometrium and the future migration into spiral arteries [106,107,108].

After blastocyst contacts with the maternal endometrium, endometrial stromal cells experience a substantial morpho-functional remodeling (decidualization), which facilitates the placentation and hemodynamic changes [109]. Almada and colleagues recently found that AEA reached very low concentrations in decidualizing endometrial cells [110], supporting the hypothesis that a low AEA tone might be important to trigger these phenotypical changes. These in vitro findings are consistent with the oscillations in plasma levels of AEA and metabolic enzyme expression reported in humans during the menstrual cycle, with lowest AEA levels occurring in the receptive mid-luteal phase [80,82].

Moreover, sustained AEA signaling shows anti-proliferative and pro-apoptotic activity in human [111] and rat [112,113] stromal cells primarily through CB1-dependent mechanisms. AEA also interferes with the human and rat endometrial stromal-derived cell viability and differentiation, increasing ceramide activity [113], modulating COX-2 dependent pathways [52,110,114,115], and affecting estradiol-mediated signaling through an anti-aromatase activity [116]. The latter is particularly relevant given that, besides ovarian estrogen, de novo synthesis of estradiol by aromatase in the uterus facilitates the stromal cell decidualization and angiogenesis processes [117]. Finally, CB1 receptor activation by AEA [110,111] or by the synthetic cannabinoid agonist WIN-55,212 [118] decreases the expression of prolactin (PRL) and insulin-like growth factor binding protein-1 (IGFBP-1) in endometrial stromal cells, partly through intracellular cyclic adenosine monophosphate (cAMP)-dependent mechanisms, confirming the negative role of CB1-mediated signaling in decidualization.

Nevertheless, CB1 deficiency in mice impairs decidualization, vascular remodeling, and formation of avascular primary decidua zone [119], highlighting that appropriate CB1 signaling has to be strictly regulated for pregnancy success.

Exogenous cannabinoids might alter CB receptor-mediated signaling leading to potentially adverse effects on early pregnancy processes. In line with this possibility, it is interesting to note that recent reports have shown that synthetic cannabinoids and ∆9-THC can impair endometrial cell decidualization trough CB1 cannabinoid receptors [118,120,121]. Furthermore, it has been reported that ∆9-THC and CBD might enhance AEA levels both inhibiting directly FAAH and modulating the activity of Fatty acid-binding proteins (FABPs), intracellular carriers that deliver AEA to FAAH for hydrolysis [122].

Finally, even more relevant in this context is the potential anti-estrogenic effect of CBD, which seems to prevent the increase in CYP19A1 gene expression and aromatase activity in differentiating endometrial stromal cells [123]. Surprisingly, ∆9-THC did not show significant anti-aromatase activity in these cells [123], whereas it disrupted placenta steroidogenesis later during pregnancy [124], confirming that different *Cannabis* constituents can modulate different signaling pathways.

Although a large part of the scientific literature has focused on AEA, a similar role during implantation and decidualization is performed by 2-AG despite its contribution to a successful pregnancy has been less investigated [74,125].

Together, these findings suggest that a low endocannabinoid signaling in the activated blastocyst and the endometrial tissues at the time of implantation is a prerequisite essential to synchronize and direct the successful early pregnancy events in healthy conditions.

### 4.2. Endocannabinoids Regulate Placentation

Endocannabinoid signaling during pregnancy has been reported to modulate the human cytotrophoblast proliferation, apoptosis, and activities required for the establishment of the proper placenta architecture. Cytotrophoblasts differentiate and fuse to form the placental syncytiotrophoblast, which represents a protective and endocrine tissue secreting human chronic gonadotropin (hCG), estrogen, human placental lactogen, and leptin and placental growth factors needed to sustain pregnancy [126]. The syncytialization is regulated by numerous factors, whose altered expression can lead to intrauterine growth restriction, preeclampsia, and other pathological conditions [126].

In vivo and in vitro studies demonstrated that 2-AG and AEA disrupt biochemical differentiation of cytotrophoblasts [127,128] and promote anti-proliferative and pro-apoptotic activities [129,130,131,132,133].

Similar results were also recently found by studying the effects of phytocannabinoids and synthetic cannabinoid agonists in two well-accepted in vitro model systems: choriocarcinoma-derived BeWo cells, which model human placental cytotrophoblasts, and HTR-8/SVneo cells, a model of first trimester human EVTs. Indeed, CBD [134], ∆9-THC [135], and synthetic cannabinoid agonists, such as WIN-55,212 [136], JWH-018, JWH-122, and UR-144 [135], have been reported to disrupt cell cycle progression and induce apoptotic cell death in BeWo cells through CB1/CB2-dependent and independent mechanisms. On the other hand, ∆9-THC exposure has been shown to attenuate proliferation, syncytialization, and mitochondrial respiration without affecting cell viability of BeWo cells [137,138,139], although it decreases migration of HTR-8/SVneo cells [139], as also observed following treatment with CBD [134].

In addition to promoting apoptotic cell death, endocannabinoids, specially 2-AG, reduce the secretion of hCG, the expression of leptin, and reduce the activity of phosphatase alkaline, impairing cytotrophoblast differentiation through cannabinoid receptor-dependent mechanisms [127], an effect also reported after ∆9-THC treatment in BeWo cells [137].

Finally, AEA participates in the regulation of maternal-fetal oxygen and micronutrient exchange in a time and dose-dependent manner. For instance, acute, but not chronic, treatment of BeWo cells with AEA decreases the transport of folic acid (FA and Vitamin B_9_), whereas, surprisingly, ∆9-THC showed an opposite stimulatory effect on FA uptake [140].

Moreover, a recent study demonstrated that treatment with AEA downregulates BCRP/ABCG2 expression and efflux activity in human placental explants and BeWo cells through CB2-mediated inhibition of cAMP synthesis [141]. Intriguingly, in vitro and in vivo studies found that CBD, Δ9-THC and other minor phytocannabinoids might act both as substrates and inhibitors of BCRP/ABCG2 [142,143,144]. In particular, Anderson and colleagues have recently shown a potential pharmacokinetic interaction of different *Cannabis* constituents at the BCRP/ABCG2 transporter located in the intestine, disclosing a potential mechanism by which one or more phytocannabinoids can enhance their plasma concentrations and physiological effects when administered in the form of full-spectrum *Cannabis* extracts compared to those reported after administration of individual constituents at equivalent doses [144]. If this phenomenon occurs at the maternal–fetal interface is still unclear, but we cannot exclude the possibility that some phytocannabinoids could have a higher local relevance than previously suspected and that *Cannabis* may affect placental permeability towards other phytochemicals and toxicants, especially in case of polysubstance abuse by pregnant women. Being that the BCRP/ABCG2 efflux transporter is highly expressed on the apical surface of syncytiotrophoblasts, where it modulates the trafficking of toxicants from maternal circulation to the developing fetus during prenatal growth [145], its downregulation could expose the fetus to detrimental effects of the xenobiotics, including *Cannabis* constituents.

In line with this scenario, treatment with ∆9-THC (3 mg/Kg, i.p.) in pregnant rats from GD 6.5 to GD 19.5 results in labyrinth specific-vascular defects and in the altered expression of placental trophoblast glucocorticoid receptors and glucose transporter 1 [146]. This uteroplacental vascularization insufficiency, which associates with fetal growth restriction, could be partially explained by the ability of AEA and 2-AG to modulate the expression of angiogenic factors, vascular endothelial growth factors and matrix metalloproteinases, as recently confirmed in the placental HTR-8/SVneo cells [108]. Accordingly, Chang and colleagues reported an impaired placental vasculature in pregnant women who smoked *Cannabis* preparations, without tobacco and alcohol use, and a defective placental angiogenesis in pregnant mice after daily treatment with ∆9-THC (5 mg/kg, i.p.) from GD 5.5 to GD 18.5 [147].

These findings in rodents are supported by a recent study on the effects of chronic prenatal cannabinoid exposure on placenta structure and functions in a translational rhesus macaque model [148]. In this study, ∆9-THC was administered in an edible form for 4 months before conception with incremental dosing until reaching 2.5 mg/7 kg/day, then maintained throughout pregnancy. Reduced placenta perfusion and oxygenation were found in ∆9-THC-exposed pregnant animals, together with increased microinfarction rates and reduced amniotic fluid volume, indicative of a dysfunctional state of the placenta. The placental insufficiency was also associated with wide-scale alterations of transcriptional profiling as assessed by RNA-sequencing, which primarily involved pathways related to angiogenesis and vascular development [148].

Together, these data confirm a key role of endocannabinoid signaling in regulating fetal cytotrophoblast differentiation, uteroplacental vascularization, and micronutrient exchange across the maternal–fetal interface. These processes depend on a low endocannabinoid tone throughout pregnancy. At the end of gestation, AEA levels significantly increase and direct the timing for onset of labor and parturition, likely by hormone-dependent mechanisms, as recently discussed by Kozakiewicz and colleagues [149].

In conclusion, the data discussed here support the hypothesis that the ECS, interacting with other local mediators, such as sex hormones, performs a complex role in a successful pregnancy. In line with this possibility, a growing number of preclinical and clinical studies have reported detrimental effects of *Cannabis*, or its constituents, on most phases of gestation.

Even more interesting is the role that the ECS and *Cannabis* might perform in the immune adaptations occurring during pregnancy, both in physiological and under inflammatory conditions. However, to date, a limited number of studies has addressed this topic. Appendix A summarizes previous studies about the role of the ECS on early pregnancy processes.

## 5. The Role of Endocannabinoids in Low-Grade Inflammation and Maternal Immune Tolerance

All immune cells express CB1 and CB2 cannabinoid receptors, and all metabolic enzymes required for the biosynthesis and degradation of endocannabinoids, as confirmed both in cell and murine models (for a review on this topic, see [150]). The expression of cannabinoid receptors significantly varies between immune cell subtypes, with the highest levels in B lymphocytes, followed by natural killer (NK) cells, monocyte/macrophages, and T lymphocytes [151]. Moreover, immune cells express other non-canonical cannabinoid receptors, such as TRPV1, GPR55, and PPARα and γ [150].

Therefore, the modulatory effects of ECS on fertilization, blastocyst implantation, placentation, labor, and parturition might be partially mediated by the regulation of the innate and adaptive immune response at the interface between embryo and decidua, enriched in immune cells [152]. Indeed, the maternal decidua hosts numerous populations of immune cells, such as decidual NK cells, macrophages, dendritic cells, cytotoxic, regulatory and helper T lymphocytes, and B lymphocytes, that dynamically change in a gestational stage-dependent manner, and show distinct signatures from their counterparts in other tissues or in the bloodstreams [153,154]. These immune cells represent the first line of defence against the pathogen colonization of the placenta [155] and are responsible for the establishment of maternal immune tolerance, a modulation of the immune response necessary to avoid the allogeneic risk due to the foreign paternal antigens expressed on the fetal tissues [156]. In physiological conditions, upon seminal fluid contact and blastocyst implantation, female tissues initiate an acute and regulated inflammatory response thought to be beneficial for pregnancy [157]. A massive infiltration of immune cells characterizes this initial inflammatory reaction, followed by a shift to an anti-inflammatory profile efficiently maintained during pregnancy through modulatory mechanisms primarily mediated by maternal regulatory (reg) T lymphocytes, decidual NK cells, and M2-like decidual macrophages [153,157,158]. These immune cells secrete many factors that stimulate the proliferation and invasion of fetal trophoblasts in the endometrium. In turn, fetal trophoblasts, together with maternal decidualizing stromal cells, promote the tolerogenic dendritic cell differentiation, M2-like macrophages polarization, CD4+FOXP3+ Treg expansion, and secretion of a plethora of immunosuppressive molecules that sustain an intrauterine tolerogenic microenvironment, including granulocyte-macrophage colony-stimulating factor (G-CSF), interleukin (IL)-10, and tumor growth factor (TGF) β [159,160,161].

### 5.1. Cannabinoids and T Lymphocytes

T lymphocytes (or T cells) are adaptive immune cells, comprising several functional subgroups, each with distinct signatures, including memory T cells, CD8+ cytotoxic T cells, CD4+ regulatory T cells, and effector CD4+ T helper (Th) cells, of which, three main subtypes are currently known: Th1, Th2, and Th17 cells [162]. At the periimplantation time, CD4+ Th1 cells are infiltrated in the maternal decidua where they regulate the trophoblast invasion of endometrium promoting vascular remodeling and angiogenesis by a pro-inflammatory response [162]. Progressively, the density of Th1 cells decreases in favor of Th2 cells, which become predominant up to parturition. CD4+ Th2 cells and Treg cells are the main responsible for the allograft tolerance during pregnancy promoting an anti-inflammatory state until the onset of a parturition-associated inflammatory reaction, which contributes to uterine contraction and delivery at the end of pregnancy [163]. Therefore, appropriate balance and temporal dynamics between different T lymphocyte subtypes at the maternal–fetal interface from periconception to parturition appear to be essential for a healthy pregnancy. Several decidual immune types, such as macrophages and NK cells, also participate to induce and maintain the Th1/Th2 ratio during pregnancy. Perturbations of their relative density and activity can adversely impact fertility and pregnancy health. For example, an excess of Th1/Th17 subtype cells has been reported in women with preterm birth [164] and recurrent miscarriage [165]. Of interest, both these pathological conditions are characterized by inflammation and high endocannabinoid levels [166,167]. It has been found that cytokines can influence endocannabinoid tone modulating FAAH enzymatic activity in human lymphocytes. In particular, anti-inflammatory Th2 cytokines, such as IL-4 and IL-10, promote FAAH catalysis, whereas pro-inflammatory Th1-derived mediators (IL-12, IFNγ) were shown to exert inhibitory effects, increasing AEA tone [168].

Interestingly, it is known that phytocannabinoids, such as ∆9-THC, can exert significant anti-proliferative and immunosuppressive functions on peripheral T lymphocytes, regulating the CD8+ and CD4+ cell balance, and cytokine production [169,170]. Part of these functions is mediated by CB2-signaling, whose stimulation was found to drive T lymphocyte differentiation toward Treg phenotype in a mouse model of inflammatory bowel disease [171]. More recently, Angelina and colleagues showed that the synthetic cannabinoid agonist WIN55,212–2 promotes human functional FOXP3+Treg cell expansion during inflammation by inducing tolerogenic dendritic cells via autophagy and metabolic reprogramming [172]. A better understanding of the mechanisms by which endo- and exo-genous cannabinoids regulate the Th1/Th2 ratio, Treg cell expansion, and the interplay with other local immune cells, especially at the maternal–fetal interface, is needed and this knowledge may help to develop novel strategies for the treatment of inflammation during pregnancy.

### 5.2. Cannabinoids and NK Cells

Decidual natural killer (NK) cells are the dominant immune cell population in the decidua during early pregnancy, where they contribute to immune tolerance, tissue remodeling, angiogenesis, and trophoblast invasion, interacting with EVT, stromal cells, macrophages, and T lymphocytes [173]. Unlike peripheral NK cells, decidual NK cells show lower cytotoxicity, and produce proangiogenic factors and chemoattractants for EVTs. The number of NK cells increases during the luteal phase of the menstrual cycle, further increases in early gestation, then progressively decreases before term [153], despite there are some conflicting data [174]. The recruitment, proliferation, and functions of NK cells in the uterus are partly regulated by ovarian-derived hormones, such as progesterone and estrogens [175]. Progesterone is also the major hormone involved in decidualization, regulating the expression of decidual markers in differentiating endometrial stromal cells, such PRL and IGFBP-1 [176,177].

Interestingly, Fonseca and colleagues recently found that the decidual NK cell-conditioned medium from women suffering from miscarriage induced a significant increase in AEA production in decidualizing endometrial stromal cells and, at the same time, downregulated cAMP-stimulated PRL and IGFBP-1 production [178].

In line with these findings, other studies have shown that treatment of human endometrial stomal cells with AEA [111] or the synthetic cannabinoid agonist WIN-55,212 [118] decreases PRL and IGFBP-1 expression via CB1-dependent mechanisms. The authors hypothesized that NK-induced increase in AEA and its effects on the decidualization of stromal cells were promoted by the pro-inflammatory state characteristics of women suffering from miscarriage, specially by tumor necrosis factor alpha (TNFα), whose levels were found upregulated in the decidual NK cells-conditioned medium derived from miscarriage samples [178]. Together, these data suggest that endocannabinoids are possible mediators for the regulation of decidualization of endometrium by NK cells under inflammatory conditions. Further investigations are needed to support this possibility and to understand if and how the ECS interferes with the several functions of this essential immune cell population.

### 5.3. Cannabinoids and Decidual Macrophages

Decidual macrophages are specialized innate immune cells that reside at the maternal–fetal interface, where they participate in the phagocytosis of dying invading trophoblasts and senescent stromal cells, fetal-antigen presentation, angiogenesis, and inflammation/infection response, as extensively studied and reviewed elsewhere [179,180,181]. Their local density varies during the menstrual cycle and the different stages of pregnancy. In particular, the number of macrophages increases from the follicular to the secretory phase of the menstrual cycle and continues to increase during the first trimester of pregnancy, representing the second more abundant decidual leukocyte population after NK cells in the early pregnancy [153,182]. From the functional point of view, decidual macrophages show high phenotypical and functional plasticity during pregnancy [180]. These cells progressively shift from an early pro-inflammatory M1-like phenotype towards a more immunosuppressive M2-like state under the influence of local stromal cells and invading trophoblasts to sustain maternal-fetal tolerance once implantation is complete [158]. Unlike M1-like macrophages that promote a pro-inflammatory immune response, M2-like macrophages favor a Th2 and Treg bias in CD4+ T cells [183] and attenuate the decidual NK cytotoxicity [184]. Interestingly, it has been reported that macrophages accumulate rapidly in the endometrium at the periimplantation window of the menstrual cycle, suggesting that these hematopoietic cells perform a role in endometrial receptivity [185]. After blastocyst implantation, decidual M1-like macrophages progressively migrate away from the implantation chambers, reducing the risk of inflammatory response towards the fetal semi-allogeneic tissues [186]. It remains unclear by which mediators the uterus regulates immune cell infiltration and migration in the primary decidual zone encircling the implantation chamber. Some evidence suggests that this process might be regulated by the ECS, whose components are entirely expressed by macrophages [150]. In fact, Li and colleagues recently found a significant retention of macrophages in the primary decidual zone of pregnant double CB1^−/−^CB2^−/−^ mutant mice on GD 6, encircling the implantation chamber. The phenotype observed in double mutant mice was also recapitulated in CB1 deficient dams, but not in CB2^−/−^ mice, indicating an important role for CB1-mediated signaling in the local macrophage recruitment during early pregnancy [119].

However, how the multifaceted regulation of macrophage biology is partly mediated by endocannabinoids remains elusive. In this sense, a potential mechanism of action by which endocannabinoids regulate macrophage activity during early pregnancy could be linked with their inhibition of adenylate cyclase activity, and secondarily of cAMP signaling [187].

Some recent studies have also highlighted a potential role of T cell immunoglobulin and mucin (TIM)-3 signaling, a transmembrane surface protein identified in Th1 cells [188], decidual macrophages [183], and NK cells [189], in the regulation of both innate and adaptive immunity. In particular, it was shown that EVT-induced TIM-3+CD14+ macrophages can promote the Th2 bias and Treg expansion in CD4+ T lymphocytes at the maternal–fetal interface, favoring immune tolerance in normal pregnancy [183]. On the contrary, in women suffering from miscarriage, TIM-3 expression was found downregulated and associated with pro-inflammatory conditions [183,188,189].

Intriguingly, there is evidence that endocannabinoids can regulate the TIM-3 signaling pathway by CB2 receptors in the microglia [190]. In addition, Yun and colleagues have shown that TIM-3 expression was increased by cAMP/PKA-dependent signaling in the human Jurkat T cell line [191].

Although these findings refer to pathological conditions and were obtained in models different from those that can be used to study the maternal–fetal interface during pregnancy, we cannot automatically exclude that similar pathways might mediate the effects of endocannabinoids on the gestational dynamics. Further studies are needed to confirm this hypothesis.

Furthermore, it was demonstrated that endocannabinoids or synthetic and natural cannabinoid compounds promote an anti-inflammatory activity in central and peripheral macrophages [152,192,193], disclosing a complex immunomodulation by these pleiotropic bioactive lipid messengers.

Overall, these findings suggest that endocannabinoids are lipidic signals that regulate, partly via cAMP, the endometrial decidualization in conjunction with progesterone and PGE, acting not only on stromal cells and trophoblasts, but also on local immune cells. Endocannabinoids might also modulate endometrial stromal cell differentiation and the ability of local NK cells, macrophages, and T lymphocytes to assist angiogenesis, spiral artery remodeling, and immune tolerance necessary to permit blastocyst development, and the term of pregnancy. Future studies should aim to clarify the role of endocannabinoids, and at which concentrations and by which pathways these lipid mediators promote or impair the adaptive and innate response during pregnancy [152]. The organ-on-chip technologies and other in vivo-like in vitro models might offer new tools to shed light on the role of endocannabinoids and the effects of *Cannabis* on the intricate crosstalk among the numerous and highly specialized cell populations residing in the maternal–fetal interface.

### 5.4. Endocannabinoids Regulate Nitric Oxide System in the Intrauterine Microenvironment: Insights from Inflammatory Conditions

Nitric oxide (NO) is a lipid-soluble gaseous free radical produced by the oxidation of L-arginine to L-citrulline in a reaction catalyzed by NO synthase (NOS). This mediator is the main vasodilator agent in the placenta, where it regulates the hemodynamic flow in addition to virtually any gestational process, including oogenesis, ovulation, implantation, placentation, uterine muscle contractility, and parturition [194]. Three main NOS isoforms have been identified in the uterus: the constitutional neuronal NOS (nNOS), the calcium/calmodulin dependent endothelial (eNOS), and the inducible NOS (iNOS) isoforms. The eNOS and iNOS isoforms are highly regulated in the implantation sites of the receptive endometrium of rodents, where NO was shown to affect the expression of the metabolic enzymes of AEA, reducing the NAPE-PLD/FAAH ratio, and, consequently, dampening endocannabinoid tone [195]. In turn, endocannabinoids were also shown to modulate NOS activity and NO production in murine decidua in a manner dependent on blastocyst presence [196]. Therefore, the NO system might interact with endocannabinoids to promote endometrial receptivity and pregnancy progression in physiological condition. Moreover, being NO a key regulator of inflammation [197], a mutual interplay between these two systems might be relevant under inflammatory conditions, as previously suggested [198]. In this respect, iNOS is predominantly expressed by M1-like decidua macrophages, which over-produce NO and nitric-reactive species in response to pro-inflammatory conditions, such as acute atherosis [199], endometriosis [200], preeclampsia [201], and in lipopolysaccharide (LPS)-induced maternal immune activation [202]. In particular, systemic administration of LPS to pregnant mice reduces plasma level of progesterone and it induces a significant increase in NOS activity and iNOS+ macrophage infiltration in the maternal decidua, leading to a high rate of embryo resorption [202,203,204,205]. Additionally, LPS exposure increases plasma levels of N-acylethanolamines in pregnant mice, including AEA, PEA, and OEA, in a progesterone-dependent manner [205]. LPS was also found to enhance AEA levels by inhibiting FAAH activity in human and rat macrophages [206]. Therefore, it is possible that LPS stimulation reduces progesterone levels rescuing the AEA tone that, in turn, promotes NO production. Indeed, reduced progesterone signaling in the periimplantation period was shown to promote implantation failure and resorption, together with an increased Th1 differentiation and reduced anti-inflammatory Th2-cytokine secretion in progesterone-induced blocking factor (PIBF)-deficient pregnant mice [207]. Moreover, no modifications in decidual NOS activity and a weaker inflammatory response to LPS were observed in CB1^-/-^ mice when compared to WT controls [205], suggesting a CB1-dependent modulation of the LPS-induced NO synthesis under inflammatory conditions [208]. A similar role for CB1 was found by Aban and colleagues, who demonstrated that pharmacological inhibition of CB1 receptors reduces the stimulatory effect induced by AEA on the NOS activity in normal and preeclamptic tissues [102].

In addition, Bariani and colleagues (2017) showed that in vivo treatment of pregnant Balb/c mice with LPS led to an increase in pro-inflammatory mediators, including iNOS, COX-2, and PGE2 and altered endocannabinoid tone [209]. Finally, endocannabinoids are involved in the premature decidual senescence following endotoxin exposure, which was associated with high risk of inflammation-induced preterm birth [166]. Together, these experimental findings support a complex interplay between the ECS and the NO system, offering another potential mechanism through which endocannabinoids regulate the intrauterine microenvironment, and in particular hemodynamic flux and the inflammatory state [210]. However, if phytocannabinoids alter this dynamic process remains unclear and needs to be further investigated. It has been shown that chronic administration of ∆9-THC before LPS exposure at GD 15 significantly reduced LPS-induced preterm births and increased gestational duration, an effect reversed by the CB1 antagonist AM281, and NOS inhibitor administration [211]. ∆9-THC and CBD were also found to be effective in alleviating the LPS-induced cytokine storm in human macrophages trough the modulation of NPRL3 inflammasome and STAT3 signaling [212]. The apparent inconsistency in the effects mediated by exogenous and endogenous cannabinoids during pregnancy might depend on their relative tissue concentration and/or involve different biological pathways [210].

## 6. From Parent to Offspring: When *Cannabis* Threatens the Neurodevelopment Trajectories

In the previous sections, we have provided evidence that *Cannabis* use during pregnancy might have significant negative effects on proper placental development by affecting several maternal- and fetal-origin cell populations. Therefore, the increased popularity of *Cannabis* use for treating symptoms associated with pre- and postnatal distress by pregnant women might represent a serious challenge to both a successful pregnancy and the offspring’s health.

Several studies suggest that maternal *Cannabis* use during pregnancy might result in adverse neonatal outcomes, including spontaneous preterm birth, fetal growth restriction, low birth weight, and more frequent intensive neonatal care unit admissions [27,213,214,215,216].

Moreover, phytocannabinoids can easily cross the BBB and directly target the fetal ECS, potentially affecting neurotransmission, synaptogenesis, and microglia activity in the offspring’s developing brain [217,218,219,220,221,222,223,224,225]. These neurochemical alterations might contribute to the higher susceptibility to develop neurodevelopmental and psychiatric disorders in the children of mothers who used *Cannabis* during pregnancy [28,29,222,224]. This conclusion is supported by many studies performed in animal models, in which a wide spectrum of behavioral and neural alterations have been reported in the offspring of dams prenatally exposed to *Cannabis* [226,227,228,229,230,231,232,233,234,235,236,237,238,239]. Interestingly, Rompala and colleagues recently found that emotional dysregulations in children whose mothers were *Cannabis* users are associated with increased stress hormone levels in the hair and, intriguingly, reduced immune-related gene expression in the placenta, suggesting that the atypical behavioral traits induced by prenatal *Cannabis* exposure might be partly linked to the immunosuppressive effects of cannabinoids [240].

Multiple biological mechanisms are hypothesized to participate to the vertical transmission of *Cannabis* effects from mother to fetus (Figure 2). Some of these effects might be consequent to placenta dysfunctions, others linked to the immunomodulatory role of phytocannabinoids at the maternal–fetal interface or may be associated with direct action of cannabinoids on the fetal developing tissues. Moreover, *Cannabis* exposure might impact epigenetic mechanisms occurring in key reproductive and brain tissues during sensitive windows of development (e.g., gestation and adolescence) [241]. Consistent with this view, several studies both in humans and rodents have begun to reveal the long-term impact of plant-derived or synthetic cannabinoids in both parental reproductive physiology and offspring’ neurodevelopment, focusing on epigenetic modifications [6].

Thus, it was recently found that ∆9-THC exposure before conception is associated with widespread changes in DNA methylome in rat sperm (F0) [242,243,244], including changes in genes important for neurodevelopment and synaptic plasticity [244,245]. Some of these epigenetic changes persist following drug cessation [243] and they are inherited through the germline from parents to offspring [243,244].

In line with these observations, additional studies have shown that periconceptional exposure of female and/or male rats (F0) to plant-derived (e.g., ∆9-THC) or synthetic (e.g., WIN55,212) cannabinoids increases the vulnerability to stress [246] and induces long lasting neurobehavioral changes in the offspring (F1) [229]. These neurobehavioral alterations were associated with altered global DNA methylation profiles in the prefrontal cortex [246] and striatum [247]. DNA and histone methylation changes associated with detrimental behavioral outcomes have also been observed in the offspring born from dams prenatally exposed to natural cannabinoids [219,235,248,249,250]. Finally, Innocenzi and coworkers have also shown that paternal chronic exposure to JWH-133, a highly selective CB2 receptor agonist, before conception affects spermatogenesis in rats and alters placenta and embryonic development [251]. These defects were associated with altered DNA methylation profiles at imprinted genes, e.g., Peg10 and Plagl1, in sperm from JWH-133 exposed males, which were conserved in the placenta after fertilization [251].

Despite evidence in humans is more limited, preliminary studies demonstrated that *Cannabis* is able to induce significant changes in spermatic count and sperm DNA methylation profile, involving genes with an important role in development [242,252]. Drug abstinence for at least one spermatogenic cycle allows partial neutralization of the *Cannabis*-associated methylation changes in sperm [252].

What still needs to be clarified is whether the maternal gamete epigenome is affected by periconceptional *Cannabis* exposure. Currently, there is evidence that endocannabinoids and ∆9-THC impact ovarian morphology, folliculogenesis, and oocyte maturation [253,254,255]. Moreover, some recent studies showed that phytocannabinoids significantly alter the global DNA methylation in human granulosa cells [256,257]. Therefore, it cannot be excluded that *Cannabis* affects the oocyte epigenome in the ovarian niche, although further studies are needed to confirm this possibility.

Together, these studies provide compelling evidence about the ability of cannabinoids not only to affect the brain and reproductive physiology of the exposed generation, but also potentially impact the epigenetic signatures and neurobehavioral functions in the F1 generation [6]. More research is needed to better understand the potential vulnerability of specific group of genes to *Cannabis* in male and female gametes, if these changes are inherited in the future generations, even in those that are not exposed to *Cannabis* directly or by exposure of germ cells, and if different cannabinoids induce distinct pattern of epigenetic signatures, potentially affected by other environmental stimuli.

## 7. Conclusions 

We have summarized the main clinical and preclinical data available on the effects of prenatal exposure to cannabinoids at the maternal–fetal interface. Due to the potential *Cannabis*-induced neurodevelopment defects in children can be detected only after birth, often in adolescence—too late to prevent potentially lifelong dysfunctions—a better understanding of the impact of prenatal *Cannabis* should remain an important focus of medical research. Currently, further studies are needed to evaluate the long-term effects associated with the use of *Cannabis*-derived preparations by pregnant women. The efforts should be directed toward the understanding of the multiple mechanisms by which endocannabinoids and *Cannabis* constituents regulate placenta and fetal development. In this regard, the use of innovative organ-on-chips technologies to model the microarchitecture and functions of maternal–fetal interface could provide interesting opportunities to study the pharmacokinetics and the effects of cannabinoids at this vital organ, their impact on different cellular maternal and fetal populations, both in physiological and inflammatory conditions, and will help the development of new and more personalized therapeutic strategies to counteract the deleterious effects of prenatal *Cannabis* exposure. Considering the increased accessibility, social acceptability, and legalization of *Cannabis* use, more research in this field will help physicians, healthcare organizations, and governments to make evidence-based decisions to safeguard the general population health.

## Figures and Tables

**Figure 1 ijms-24-05220-f001:**
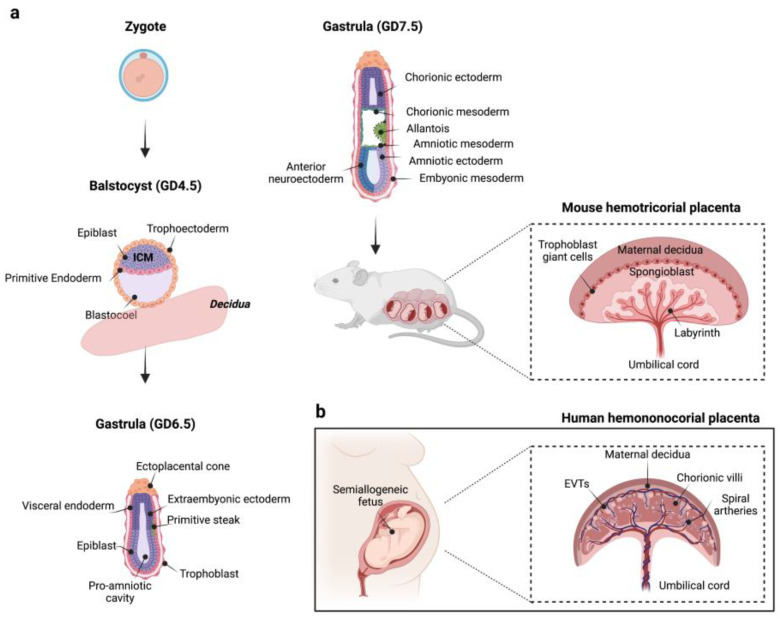
(**a**) A schematic representation of mouse placenta development. At gestational day (GD) 3.5–5.0, the mouse blastocyst is composed by ICM, which contains precursor cells for embryonic tissues (epiblast), and visceral and parietal endoderm (primitive endoderm), and by an outer cell layer called trophectoderm. Starting from GD 5.0–6.5, the polar trophectoderm differentiates in the ectoplacental cone, which gives origin to invading trophoblasts and junctional zone, and in the extraembryonic trophectoderm, which, in turn, gives rise to the chorionic ectoderm (GD 6.5–9.5). At the same time, the mesoderm (primitive streak) emerges between visceral endoderm and ectoderm tissues. Subsequently, the chorion fuses with mesodermal-derived allantois promoting placental labyrinth formation. As pregnancy progresses, a definitive placental structure becomes evident around the second week of gestation (GD 10–14.5). (**b**) A schematic representation of human placenta architecture. Figure created with “https://www.biorender.com/ (accessed on 18 January 2023)”.

**Figure 2 ijms-24-05220-f002:**
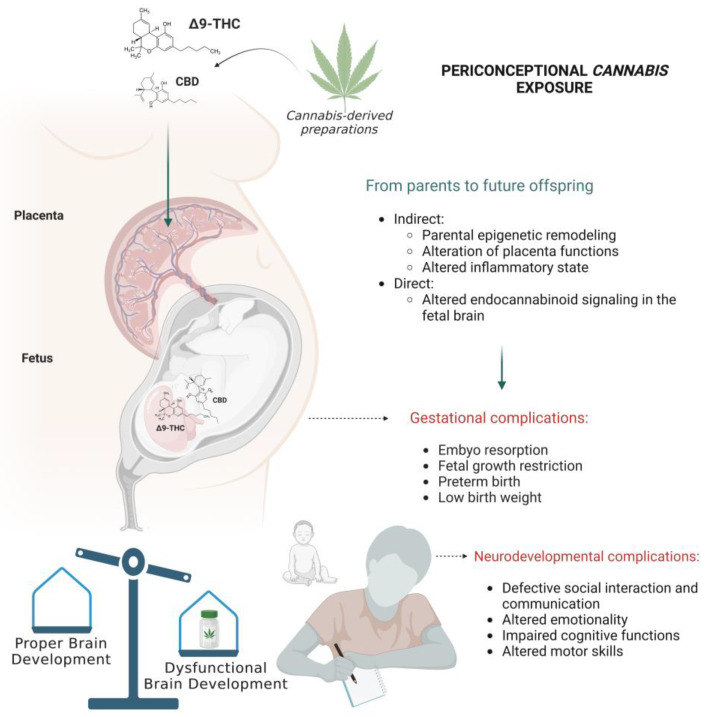
Prenatal *Cannabis* exposure can affect gestational processes and neurodevelopmental trajectories in the offspring through both direct and indirect mechanisms. Figure created with “https://www.biorender.com/ (accessed on 18 January 2023)”.

## Data Availability

Not applicable.

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
