# Peer review of "Emerging Roles of Endocannabinoids as Key Lipid Mediators for a Successful Pregnancy"

_ijms, 2023, doi:10.3390/ijms24065220_

Round 1

Reviewer 1 Report

The authors discuss the impact of the administration of cannabinoids during pregnancy. However, the text is sometimes not son well organized so it is not clear whether there are explaining the effects on the mother or the fetus. Moreover, since they are specifically discussing the effects on the immunotolerance, they should conduct the approach of the introduction towards this topic as well, to make clear first what is happening under physiological conditions. Concerning the transgenerational effects some mistakes must be corrected.

Line 93 please replace targets by receptors

Line 211 the structures mentioned by the authors are not all tissues so please correct it

Lines 247-259: the role of CB1 in pregnancy is a matter of discussion, especially for during trophoblast migration. Authors should deeply develop this matter not only in mice models but also in humans since changes in this receptor have been linked to different pathologies (PMID: 33511667, PMID: 36361721 , 25444073).

Figure 2 is out of the focus of the review. Authors should summarize in figures basic information to understand the main messages of the review.

Why authors have exclusively pointed to the role of cannabinoids in immune tolerance? Then it will be more accurate to change the title of the review, pointing to the role of endocannabinoids specifically in the immune changes related to pregnancy

In line 640 authors have stated transgenerational but in fact, through the text, they are not explaining the effects of cannabinoids through different generations (including the non-exposed) but only to the offspring, so the term transgenerational is not accurate here since evident change in the F3 must be reported. Moreover, taking into account the important role of epigenetics in the transgenerational effects… Is there some evidence regarding the impact of cannabinoids on DNA methylation or histone acetylation in the placenta? The information regarding this topic is very scarce in the manuscript and it could be of the utmost interest.

In line 657 why authors mean by long-lasting? Intergenerational? Transgenerational?

Author Response

Reviewer #1

Reviewer #1 (Comments to the Authors):

Reviewer’s comment 1:

The authors discuss the impact of the administration of cannabinoids during pregnancy. However, the text is sometimes not son well organized, so it is not clear whether there are explaining the effects on the mother or the fetus. Moreover, since they are specifically discussing the effects on the immunotolerance, they should conduct the approach of the introduction towards this topic as well, to make clear first what is happening under physiological conditions. Concerning the transgenerational effects some mistakes must be corrected.

Authors’ reply to Reviewer’s comment 1:

We thank the Reviewer for this comment. We have modified the text so that the maternal effects of Cannabis exposure are more easily identifiable by the reader than their fetal counterpart. Moreover, we added a table addressing this issue, as also suggested by Reviewer 2. Finally, we modified the section relative to potential epigenetic effects induced by pre-conceptional use of Cannabis, as specified later.

Reviewer’s comment 2:

Line 93 please replace targets by receptors

Authors’ reply to Reviewer’s comment 2:

We thank the reviewer for the observation. We have modified the text accordingly).

Reviewer’s comment 3:

Line 211 the structures mentioned by the authors are not all tissues so please correct it.

Authors’ reply to Reviewer’s comment 3:

We thank the reviewer for the comment. We have integrated the list of tissues where constituents of ECS have been previously described in humans or rodents (lines 206-209 of the revised manuscript).

Reviewer’s comment 4:

Lines 247-259: the role of CB1 in pregnancy is a matter of discussion, especially for during trophoblast migration. Authors should deeply develop this matter not only in mice models but also in humans since changes in this receptor have been linked to different pathologies (PMID: 33511667, PMID: 36361721, 25444073).

Authors’ reply to Reviewer’s comment 4:

We thank the Reviewer for pointing this out. We have discussed more clearly the role of CB1 receptors in the early gestational processes in humans and rodents, highlighting the inconsistency between results, where present.

Reviewer’s comment 5:

Figure 2 is out of the focus of the review. Authors should summarize in figures basic information to understand the main messages of the review.

Authors’ reply to Reviewer’s comment 5:

Following the Reviewer's comment, we removed Figure 2 from the revised Manuscript.

Reviewer’s comment 6:

Why authors have exclusively pointed to the role of cannabinoids in immune tolerance? Then it will be more accurate to change the title of the review, pointing to the role of endocannabinoids specifically in the immune changes related to pregnancy.

Authors’ reply to Reviewer’s comment 6:

We thank Reviewer for this comment. However, the present manuscript discusses the effects of endogenous and exogenous cannabinoids on development and functioning of the maternal-fetal interface, and it is not exclusively focused on their immunomodulatory roles. For this reason, we feel the current title would better convey the main message of this review. 

Reviewer’s comment 7:

In line 640 authors have stated transgenerational but in fact, through the text, they are not explaining the effects of cannabinoids through different generations (including the non-exposed) but only to the offspring, so the term transgenerational is not accurate here since evident change in the F3 must be reported. Moreover, taking into account the important role of epigenetics in the transgenerational effects… Is there some evidence regarding the impact of cannabinoids on DNA methylation or histone acetylation in the placenta? The information regarding this topic is very scarce in the manuscript and it could be of the utmost interest.

Authors’ reply to Reviewer’s comment 7:

We are grateful to the Reviewer for this comment. In effect, environmental conditions can influence more than one generation trough distinct modes. However, since evidence regarding the transgenerational effects of Cannabis beyond F1 are scarce, and in line with the Reviewer's opinion, we have decided to change the title of this paragraph in: “From parents to offspring: when Cannabis threatens the neurodevelopment trajectories”. In this section, we briefly discuss the direct and indirect mechanisms by which cannabinoids can affect the offspring’s health, describing more in detail the potential role of epigenetic changes.

Reviewer’s comment 8:

In line 657 why authors mean by long-lasting? Intergenerational? Transgenerational?

Authors’ reply to Reviewer’s comment 8:

Following the Reviewer’s comment, we have rephrased the sentence (line 654-656) as follows:

“… This conclusion is supported by many studies on animal models, in which a wide spectrum of behavioral and neural alterations have been reported in the offspring of dams prenatally exposed to Cannabis”.

Reviewer 2 Report

Emerging roles of endocannabinoids as key lipid mediators for 2 a successful pregnancy

1.     The whole manuscript needs excessive English editing.

2.     Line 10, what does it mean Keywords (3-10)

3.     Add the list of abbreviations.

4.     Some references need to be updated.

5.     The Policy of Cannabis sativa should be added clearly.

6.     The study should include a table about the past study concerning the endocannabinoids studies.

Author Response

Reviewer #2

Reviewer #2 (Comments to the Authors):

Reviewer’s comment 1:

The whole manuscript needs excessive English editing

Authors’ reply to Reviewer’s comment 1:

Following the Reviewer’s comment, we performed an extensive revision of the English language and style of the whole manuscript.

Reviewer’s comment 2:

Line 10, what does it mean Keywords (3-10)

Authors’ reply to Reviewer’s comment 2:

Thanks to the Reviewer for this observation. We have removed the range (3-10) from the keywords list at line 21. We apologize for this mistake.

Reviewer’s comment 3:

Add the list of abbreviations.

Authors’ reply to Reviewer’s comment 3:

We thank Reviewer for his comment. We have added a list of abbreviations at the end of the manuscript.

Reviewer’s comment 4:

Some references need to be updated.

Authors’ reply to Reviewer’s comment 4:

Based on the Reviewer’s comment, we reviewed and integrated the most recent literature in the text and bibliography.

Reviewer’s comment 5:

The Policy of Cannabis sativa should be added clearly.

Authors’ reply to Reviewer’s comment 5:

Following the Reviewer’s comment, in the introduction of the revised manuscript we better discussed the current policy on Cannabis use across western Countries.

Reviewer’s comment 6:

The study should include a table about the past study concerning the endocannabinoids studies.

Authors’ reply to Reviewer’s comment 6:

Following the Reviewer’s suggestion, we inserted a table (Table 1) as supplementary material summarizing the previous studies about the role of the endocannabinoid system on the early pregnancy processes.

Round 2

Reviewer 1 Report

First of all, I thank the authors for their effort in improving the manuscript. Still, I have some concerns that should be mended prior to publication. 

Authors should be aware that they have written a review should they should compilate all or most the results regarding the topic and discuss them (especially regarding the role of CB1 in placenta under physiological and pathological conditions). Moreover, authors should further explain the type of epigenetic changes since they can have a different impact on gene transcription and, therefore, on cell function. 

Line 29 countries without capital letter, and also in line 50

Line 54, please specify what do you mean with small amounts

Regarding CB1 effects related to preeclampsia, the authors should discuss deeper about it, as done for the ectopic pregnancy, since I have already suggested it in the previous review, there’s discrepancy and different results have been shown, not only the ones from Fugedi and colleagues (doi: 10.1016/j.placenta.2012.10.009 and doi: 10.3390/ijms232112931. Taking into account that this is a review manuscript authors should try to put together all the results and discuss the differences among them, otherwise some information will be missing.

Line 275 in vitro without hyphen. Same in line 325

In line 676, when authors stated “including changes in 676 genes important for neurodevelopment and synaptic plasticity” have these results been found in F0 or F1?

In line 680 please specify which type of behavioral abnormalities

In line 681 replace preconceptional by periconceptional

In line 686 please specify the type of epigenetic changes

In line 695 can authors please specify which are the epigenetic effects? And in line 696 I guess is not accurate to use “amelioration” but neutralization instead.

In line 702 please specify which type of epigenetic signatures

In figure 2 please replace prenatal by periconceptional

Author Response

Reviewer #1 (Comments to the Authors):

Reviewer’s comment 1:

Authors should be aware that they have written a review should they should compilate all or most the results regarding the topic and discuss them (especially regarding the role of CB1 in placenta under physiological and pathological conditions). Moreover, authors should further explain the type of epigenetic changes since they can have a different impact on gene transcription and, therefore, on cell function. 

Authors’ reply to Reviewer’s comment 1:

We thank the Reviewer for his comments. We modified the manuscript in line with Reviewer’s recommendations.

Reviewer’s comment 2:

Line 29 countries without capital letter, and also in line 50

Authors’ reply to Reviewer’s comment 2:

The text has been modified accordingly.

Reviewer’s comment 3:

Line 54, please specify what do you mean with small amounts

Authors’ reply to Reviewer’s comment 3:

We thank the Reviewer for his comment. We have now clarified in the text that we are referring to drug quantities which do not exceed that required for average individual consumption (line 54-55 of the revised manuscript).

Reviewer’s comment 4:

Regarding CB1 effects related to preeclampsia, the authors should discuss deeper about it, as done for the ectopic pregnancy, since I have already suggested it in the previous review, there’s discrepancy and different results have been shown, not only the ones from Fugedi and colleagues (doi: 10.1016/j.placenta.2012.10.009 and doi: 10.3390/ijms232112931. Taking into account that this is a review manuscript authors should try to put together all the results and discuss the differences among them, otherwise some information will be missing.

Authors’ reply to Reviewer’s comment 4:

Thank the Reviewer for this advice. We have better discussed the evidence about the role of CB1 in preeclampsia (lines 264-274 of the revised manuscript).

Reviewer’s comment 5:

Line 275 in vitro without hyphen. Same in line 325

Authors’ reply to Reviewer’s comment 5:

We have removed the hyphens from “in-vitro” along the whole manuscript.

Reviewer’s comment 6:

In line 676, when authors stated “including changes in 676 genes important for neurodevelopment and synaptic plasticity” have these results been found in F0 or F1?

Authors’ reply to Reviewer’s comment 6:

Following the Reviewer’s comment, we have specified that epigenetic changes are referred to F0 generation (line 692 of the revised manuscript).

Reviewer’s comment 7:

In line 680 please specify which type of behavioral abnormalities

Authors’ reply to Reviewer’s comment 7:

This sentence has been modified (line 695-696).

Reviewer’s comment 8:

In line 681 replace preconceptional by periconceptional

Authors’ reply to Reviewer’s comment 8:

As suggested by the Reviewer, we changed preconceptional with periconceptional where suggested (line 697 of the revised manuscript).

Reviewer’s comment 9:

In line 686 please specify the type of epigenetic changes

Authors’ reply to Reviewer’s comment 9:

We have specified the type of epigenetic changes indicated in the cited studies (lines 702-703 of the revised manuscript).

Reviewer’s comment 10:

In line 695 can authors please specify which are the epigenetic effects? And in line 696 I guess is not accurate to use “amelioration” but neutralization instead.

Authors’ reply to Reviewer’s comment 10:

We have revised these two sentences as follows:  

“Despite evidence in humans is more limited, preliminary studies demonstrated that Cannabis is able to induce significant changes in spermatic count and sperm DNA methylation profile, involving genes with an important role in development [242, 252]. Drug abstinence for at least one spermatogenic cycle allows partial neutralization of the Cannabis-associated methylation changes in sperm (line 711-716 of the revised manuscript).

Reviewer’s comment 11:

In line 702 please specify which type of epigenetic signatures

Authors’ reply to Reviewer’s comment 11:

We have better specified the type of epigenetic changes indicated in the cited studies (lines 720-721 in the revised manuscript).

Reviewer’s comment 12:

In figure 2 please replace prenatal by periconceptional

Authors’ reply to Reviewer’s comment 12:

Thank the reviewer for this observation. As suggested, we have changed prenatal with periconceptional in figure 2.
